# Effects of Physical Exercise on the Physical and Mental Health of Family Caregivers: A Systematic Review

**DOI:** 10.3390/healthcare13101196

**Published:** 2025-05-20

**Authors:** Ana Bravo-Vazquez, Ernesto Anarte-Lazo, Juan Jose Gonzalez-Gerez, Cleofas Rodriguez-Blanco, Carlos Bernal-Utrera

**Affiliations:** 1Doctoral Program in Health Sciences, University of Seville, 41004 Seville, Spain; abravovazquez2@gmail.com; 2Faculty of Health, UNIE University, 28015 Madrid, Spain; ernesto.anarte@universidadunie.com; 3Department Nursing, Physiotherapy and Medicine, University of Almeria, 04120 Almeria, Spain; 4Physiotherapy Department, Faculty of Nursing, Physiotherapy and Podiatry, University of Seville, 41009 Sevilla, Spain; cleofas@us.es (C.R.-B.); cbutrera@us.es (C.B.-U.)

**Keywords:** caregiving, exercise, physical activity, quality of life

## Abstract

The number of family caregivers of dependent older adults is increasing. The adverse effects of the work provided by these caregivers can have a negative impact on their own physical and mental health, so it is necessary to develop strategies that support and improve the quality of life and functional capacity of this group. **Background/Objectives**: The aim of this systematic review is to analyze physical exercise interventions for family caregivers and the effects on their physical and mental health, quality of life and functioning. **Methods**: A comprehensive search was conducted in the scientific databases PubMed, Embase, Scopus and CINAHL. Data extraction was carried out from the selected articles, obtaining information about the characteristics of the study subjects, type and characteristics of the intervention and results. **Results**: A total of 17 studies were selected for the review. All studies were based on physical exercise interventions and reported significant improvements in caregivers’ physical and mental health, as well as an increase in their quality of life and functioning. Most of the study subjects were older adult women relatives. No adverse effects were found to the interventions. **Conclusions**: Physical exercise seems to be effective in improving the physical and mental health of family caregivers, increasing their quality of life and functional capacity. More future research is needed to make interventions more accessible to family caregivers.

## 1. Introduction

In recent decades, global society has experienced a steady increase in population ageing, a trend that is expected to intensify in the coming years. By 2030, it is estimated that one in six people worldwide will be aged 60 or older, with the population in this age group rising from 1 billion to 1.4 billion. Furthermore, by 2050, this number is projected to double to 2.1 billion. The population aged 80 and over is also expected to triple between 2020 and 2050, reaching 426 million individuals [1].

This demographic shift implies a corresponding rise in the number of dependent older adults, as increased life expectancy is often accompanied by multimorbidity and functional decline [2]. The growing number of dependents poses significant economic and social challenges, particularly in light of the global shortage of healthcare professionals and limited financial resources. In this context, informal caregiving—typically provided by family members—has become essential for addressing the needs of dependent older adults. Consequently, the number of family caregivers has been steadily increasing [3,4,5].

Caring for a dependent older adult can be highly stressful and may negatively affect caregivers’ physical and mental health. Numerous studies have documented the decline in health among family caregivers [6,7,8,9,10,11]. For instance, Lambert et al., in a study involving 300 family caregivers, found that caring for a dependent individual significantly reduces caregivers’ quality of life [12]. Similarly, Pinquart and Sörensen, through a meta-analysis of 84 studies comparing caregivers and non-caregivers, reached the same conclusion [13]. In response, a wide range of interventions—primarily based on health education programs—have been developed to support caregivers and enhance their quality of life [14,15].

Also, caregivers have a higher risk of cardiovascular disease than non-caregivers, where the main modifiable risk factor is physical inactivity [16]. Physical activity is beneficial for health, as it not only reduces the risk of cardiovascular diseases, but also improves chronic diseases and reduces pain, thus increasing life expectancy [17]. The World Health Organization (WHO) defines physical exercise as: “A subcategory of physical activity that is planned, structured, repetitive and purposeful in the sense that the improvement or maintenance of one or more components of physical fitness is the objective” [18]. There are studies that show the benefits of physical exercise in patients suffering from metabolic diseases or some type of cancer. In this population, physical exercise shows improvements in general well-being, mental health, sleep quality and even functional capacity [17,19,20].

In summary, physical exercise may enhance both the physical and mental health of family caregivers of dependent older adults, thereby improving their quality of life and functional capacity, and ultimately contributing to the continuity and quality of care provided. According to the existing literature, few reviews have specifically examined the benefits of physical exercise for caregiver health [21,22]. This systematic review aims to analyze and evaluate the current evidence on the effects of physical exercise on the physical and mental health, quality of life and functional capacity of family caregivers of dependent older adults. By doing so, it seeks to expand the limited body of literature on this topic and to highlight emerging intervention strategies—such as telerehabilitation and immersive methods—that have gained traction in recent years.

## 2. Materials and Methods

This systematic literature review was conducted according to the Preferred Reporting Items for Systematic Reviews and Meta-Analysis (PRISMA) guidelines [23] and was prospectively registered on the International Prospective Register of Systematic Reviews platform, using the following registration number CRD42025645292. No financial support was provided for this study.

An advanced literature search was conducted in February 2025 in the scientific databases PubMed, Embase, Scopus and CINAHL. The terms used were “exercise”, “physical therapy modalities”, “caregivers”, “physical activity”, “physiotherapy” and “caregiver”, according to the EMTREE thesaurus of the Embase database, CINAHL Headings of CINAHL and MeSH for PubMed and Scopus. See Appendix A. Only articles that referred to family caregivers of dependent older adults were selected. Additionally, the reference lists of studies identified as eligible following the search were hand-searched to ensure that no relevant studies were missed. Grey literature was not included in our review. All items were stored in Mendeley Reference Manager.

### 2.1. Eligibility Criteria

The inclusion and exclusion criteria of the studies to be included in the review were defined by the PICOS framework [24]

#### 2.1.1. Inclusion Criteria

(P): Adult population dedicated to the unpaid care of a dependent older adult regardless of sex, age or ethnic origin.

(I): Physical exercise interventions. WHO defines physical exercise as: “*A subcategory of physical activity that is planned, structured, repetitive and purposeful in the sense that the improvement or maintenance of one or more components of physical fitness is the objective*” [18].

(C): No intervention at all or interventions based on health education programs.

(O): Physical health, mental health, quality of life and functional capacity. Studies whose results are measured by validated scales or questionnaires.

(S): Randomized controlled trials (RCTs) published in any language.

#### 2.1.2. Exclusion Criteria

We did not include reviews, studies conducted in family caregivers of children, professional caregivers, and interventions that do not involve physical activity.

### 2.2. Study Selection

Two reviewers (AB and CB) independently screened titles/abstracts against the prespecified inclusion/exclusion criteria. For those that met the inclusion criteria, the full texts were obtained. Moreover, if any uncertainty existed, the full text was retrieved for further clarification. If needed, the authors of the original work were contacted. Screening of full texts was conducted in the same manner using the predefined inclusion/exclusion criteria.

Articles were included when eligibility was confirmed by both reviewers. Any disagreement between the two reviewers was first discussed in a consensus meeting between both reviewers, and if no agreement could be made, an independent reviewer (CR) was sought to decide about inclusion/exclusion. Reviewers were not blinded to journal titles or study authors.

### 2.3. Data Extraction

Data extraction was conducted by one reviewer (AB) and then checked by a second reviewer (CB) through a data extraction sheet previously developed by consensus of all authors.

The following data were extracted from each of the selected articles: first author, year of publication and country; details of the study subjects, such as sample size, sex, age, and relationship to the dependent person; details of the intervention such as exercise modality, number and duration of sessions, and details of the control group; results and main conclusions. This information was synthesized and displayed in a table of characteristics. See Table 1. The extended table can be found in Appendix A.

### 2.4. Quality Assessment and Risk of Bias Assessment

The risk of bias was evaluated using the Cochrane RoB2 tool [42]. This tool is made up of five different domains: randomization process; deviations from planned interventions; lack of result data, measurement tools; and selection of reported findings. In each domain, it is necessary to answer several “signaling questions”. The overall risk of bias may be judged as “high” or “low” or may indicate “some concerns”.

Two independent review authors (EA and JJG) performed the analysis and applied the study scales individually. In the event of a dispute, a third reviewer (AB) would resolve the dispute. Interrater reliability was calculated through Cohen’s Kappa coefficient.

## 3. Results

### 3.1. Search Results

The search in all databases yielded a total of 272,523 articles, of which 271,984 were ineligible and 461 duplicate articles were eliminated. A total of 78 articles were analyzed by title and abstract and 30 articles were excluded because they did not meet any criteria. Finally, 48 articles were analyzed in full text and 31 articles were excluded because they met exclusion criteria. A total of 17 articles were included for the review [25,26,27,28,29,30,31,32,33,34,35,36,37,38,39,40,41]. See Figure 1.

### 3.2. Study Characteristics

All studies included in the review used RCT research designs, providing more accurate evidence. The studies were conducted on a total of 1403 subjects, with the sample size ranging from n = 31 [40] to n = 211 [26]. All studies were conducted in a higher proportion of female caregivers, and in five studies participation was exclusively female [25,27,30,33,41]. The average age range of caregivers in the 17 included articles was 50 to 74 years, studies by Lök et al. and Yilmaz et al. [31,41] indicated the lowest mean age of 50 years and Hirano et al. indicated the highest age of 74 years [40]. All participants in the 17 studies were family carers and most of them were children and spouses.

### 3.3. Type and Characteristics of Interventions

In the 17 articles included, physical activity interventions aimed at family caregivers were carried out; five of them were based on aerobic exercises [29,34,35,37,38], another study used muscle relaxation exercises [25] and another of article included guided walks [39]. Lök et al. structured the exercises in their study with a warm-up prior to the exercise routine and a cool-down afterwards [41]. There are only two authors who combined a program of physical exercises with a program of education for caregivers to improve their physical health [25,37].

In most studies, exercises were performed individually, authors such as Kim et al. included group sessions [29], and in the study by Lin et al., the exercises were performed at home, but also in a group manner via a social web application [36]. Of the 17 studies included, 10 of them were based on exercises that are performed from home [26,27,28,30,31,32,33,34,36,37]. Most studies offered phone support or motivational text messages. Only four of the studies reported monitoring of participants’ activity via pedometer [26,39,40,41].

The duration of the interventions varied, ranging from 6 weeks [36] to 12 months [26,27,33]. The frequency of most studies was two to three sessions per week, although we found interventions that employed a lower frequency of one session per week [29,41], and a higher frequency of five sessions per week [30,39], studies such as those by Farran et al. or Lin et al. simply set weekly physical activity goals [26,36]. The intensity of the interventions varied widely, although in most cases there was an increase over the course of the study. See Table 1. The extended table can be found in Appendix A.

### 3.4. Study Results

#### 3.4.1. Physical Health Results

Of the 17 included studies, there are only seven reported measures of caregivers’ overall physical health, all of which had positive results compared to control groups [25,26,29,34,35,37,40]. Farran et al. further reported improvements in strength and endurance, also measuring the number of steps of participants with the use of a pedometer [26]. Another study further targeted health outcomes by measuring hand grip strength and upper and lower limb strength and found improvements in the 6 min walk test [37]. Some authors conducted a comprehensive study of participants’ physical health, also assessing flexibility and mobility [25,34].

Montero-Cuadrado et al. used the Spanish version of the SF-36 V2, in which patients also reported improvements in pain perception [25]. Puterman et al. went even deeper by measuring, through blood tests, the participants’ telomere length and body mass index, finding positive results and improvements in cardiorespiratory exercises [35].

Other authors also conducted questionnaires and interviews to identify the perception and well-being of the subjects when performing physical exercise [32,34,39], and one of them even analyzed the motivation of caregivers to do any physical activity [33].

#### 3.4.2. Mental Health Results

Most of the studies included reported outcomes on caregivers’ mental health, all of them indicating improvement compared to control groups [25,27,28,29,30,31,32,34,36,37,38,39,40,41]. Only one of the studies concluded that physical activity is beneficial in the mental health of caregivers, except those who cared for people with cognitive impairment; in these cases, the improvement was negligible [30].

The caregiver burden was assessed in eleven of the studies, nine of them using the Zarit scale [25,28,29,30,31,34,38,40,41] and two using validated interviews and questionnaires [26,37].

Depression was measured in nine of the included studies [25,27,28,29,30,31,32,38]. Some of them used the Beck Depression Scale to measure outcomes [27,31], and others used the Geriatric Depression Scale [25,28,29,30,34]. Two of the authors, however, used validated interviews and questionnaires [32,38]. Some of the studies also reported stress level results using the Perceived Stress Scale (PSS) [27,29,32,36,39].

#### 3.4.3. Quality of Life and Functional Capacity

Quality of life and functional capacity were assessed by four of the authors, and all of them reported positive results with respect to the control group. Two of the authors used the Spanish version of the SF-36 to measure the results [25,34], and the others used the Healthy Life Style Behavior Scale [29,41].

Two of the studies also assessed sleep quality [27,40], showing significant improvements in subjects in the intervention group.

#### 3.4.4. Adverse Events

None of the studies consulted reported adverse events of interest [25,26,27,28,29,30,31,32,33,34,35,36,37,38,39,40,41].

#### 3.4.5. Dropout Rate

The average dropout rate was 11.8%, with a dropout rate ranging from 0% [39,41] to 25% [30]. Despite the low dropout rate, in most studies attributing attrition to caregiver time constraints, which could have led to some bias, as participants with the highest burden of care were those who dropped out.

#### 3.4.6. Quality of Studies and Risk of Bias Assessment

The majority of studies included in the analysis demonstrated a low risk of bias and a high level of methodological quality [25,26,28,29,30,31,34,35,36,37,38,39,41], and only three of the RCTs appeared to be at high risk of bias [27,33,40]. See Figure 2.

The interrater reliability was 82.7% (k = 0.827), which refers to a good agreement.

## 4. Discussion

The aim of this systematic review was to examine the existing evidence on the benefits of physical exercise on the physical and mental health of family caregivers of dependent older adults. A total of 17 studies were reviewed, most of which involved older adult female caregivers. The findings indicate that physical activity significantly improves both the physical and mental health of family caregivers, positively impacting their quality of life and functional capacity in the same way. Physical exercise, therefore, represents an important aid for family caregivers to improve and preserve their quality of life and functional capacity. In this way, physical exercise also helps family caregivers to prolong the provision of care to dependent people, thus contributing to solving a problem that will arise in the future since the number of dependent people is increasing and the life expectancy of this population is increasing [1,2].

Most of the reviewed studies focused on the mental health benefits of caregivers, with fewer also referring to the benefits of physical health, quality of life and functional capacity. There is therefore a need to increase the number of investigations in which the physical benefits of these interventions are also analyzed, since the physical well-being of family caregivers is also affected by the provision of care and influences their quality of life and functional capacity [6,7,8,9,10,11,12,13].

Although the benefits of physical exercise programs are well-documented, family caregivers often face numerous barriers to participation. The primary barriers or limitations are time and place, since these caregivers are unable to be away from home for a long time due to their caregiving responsibilities [43]. In this review, interventions in which physical exercise was performed at home without these barriers were analyzed [26,27,28,30,31,32,33,34,36,37].

In the study conducted by Lin et al. [30], a physical activity intervention was implemented in family caregivers through an application for mobile phones, so that time and place were no longer a limitation as each caregiver could individually adapt the place and time in which to perform the activity. The authors suggest that social support functions as a key mechanism for promoting health improvement.

Madruga et al. [25] take a closer look at how doing physical exercise at home might affect how burdened caregivers feel. As reported in their study, subjective caregiver burden represents a significant barrier to engaging in physical activity. However, the authors concluded that exercising at home can positively influence these symptoms, emphasizing that mental health challenges may hinder participation in physical exercise programs among family caregivers.

In another study, Castro et al. [32] examined adherence to home-based exercise programs among caregivers and reported encouraging results. The authors concluded that, in addition to performing physical exercises from home, constant contact with health professionals reduces stress and further increases the level of adherence to physical exercise programs. Similar findings were reported by Gary et al. [40] in the study analyzed in this review, in which they compared the results of family caregivers who performed physical exercise with family caregivers who received psychoeducation talks in addition to exercise. Their results showed that the group that combined exercises with psychoeducation obtained better health outcomes since the benefits of psychoeducation sessions in mental health increased adherence to the physical exercise intervention, and this in turn also produced benefits in mental health.

Other studies analyzed, such as that by Farran et al. [39], also relate adherence to physical exercise programs from home with socialization, concluding that the lack of socialization in this type of intervention was one of the main causes of dropout in the participants of their study.

This type of social isolation was also evident during the COVID-19 pandemic, when telemedicine and telerehabilitation emerged as practical and effective alternatives for remotely managing various health conditions [44]. In this context, online interventions delivered via videoconferencing represent a promising approach to enhance participation and adherence among family caregivers [45], offering a flexible format that can be adapted to home-based settings.

Future studies should explore the use of telerehabilitation, as it would allow family caregivers to carry out online physical exercise programs from home, without having to neglect the dependent person and adapting to schedules. In addition, telerehabilitation allows a professional to supervise the exercises and ongoing feedback between professionals and caregivers. These online exercise programs could also be delivered in groups formats, increasing the socialization of family caregivers, thus reducing the dropout rate and increasing adherence.

This review has several limitations, such as the small sample size of some of the studies analyzed. Another limitation is the dropout rate in some of the studies, which can interfere with the results, so future research should be carried out in which interventions are more accessible and enjoyable for caregivers. Another limitation is the absence of a meta-analysis, as this review was restricted to a qualitative analysis of the evidence, so as a future perspective, research should be carried out accompanied by meta-analysis.

Finally, a further limitation is that all the studies included in this review were RCTs as they provide the most accurate evidence, thus, incorporating other study designs in future reviews could provide additional insights and a more comprehensive understanding of the topic.

The main clinical implication of this review is that family caregivers of dependent older adults may enhance their quality of life and functional capacity, which would improve the quality and sustainability of the care they provide. Experts in physical exercise could implement and supervise online exercise programs for this population, thereby facilitating participation and promoting long-term adherence.

## 5. Conclusions

Physical exercise appears to be effective in improving the physical and mental health of family caregivers, as well as enhancing their quality of life and functional capacity. Future research should explore how the use of new technologies can make interventions more accessible to family caregivers, thereby increasing participation and adherence.

## Figures and Tables

**Figure 1 healthcare-13-01196-f001:**
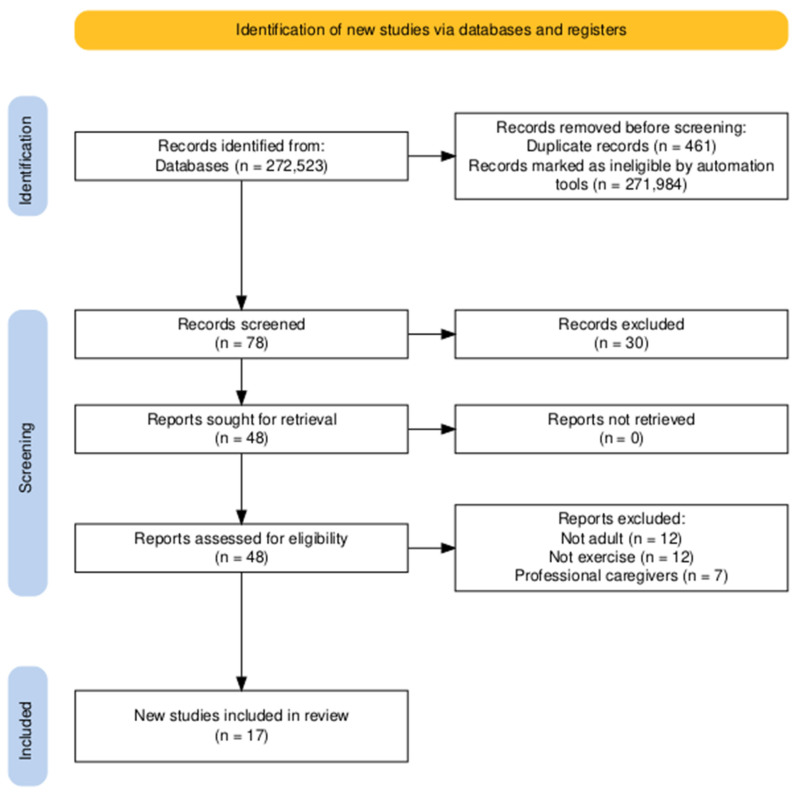
PRISMA flowchart [23].

**Figure 2 healthcare-13-01196-f002:**
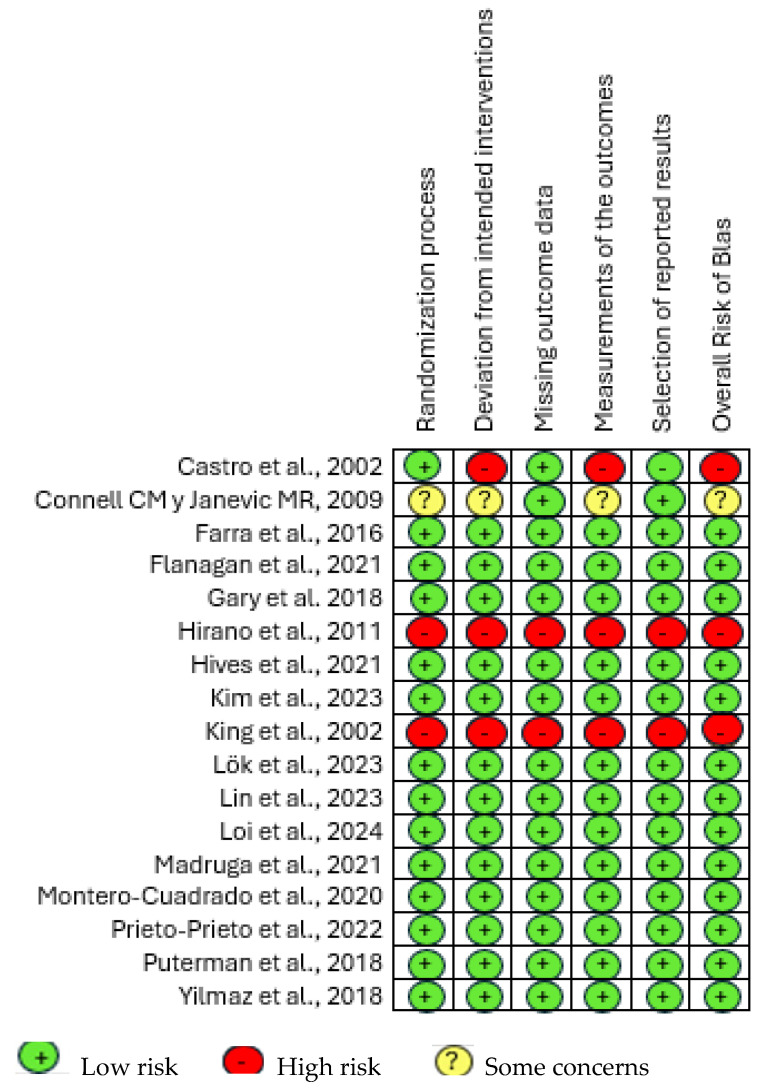
RoB Chart [25,26,27,28,29,30,31,32,33,34,35,36,37,38,39,40,41].

**Table 1 healthcare-13-01196-t001:** Summary of the study characteristics and main results.

Author(s), Year, Country	Participants Characteristics (Number, Gender, Mean Age, Relationship to Dependent)	Intervention (Type, Duration, Frequency, Intensity) and Comparison (Control Group)	Main Results
Madruga et al., 2021 Spain [25]	N = 48, female, 61 years old, wives	Intervention: home-based physical exercise program; Duration: 36 weeks; Frequency: 2 sessions/week, 60 min/session; Intensity: moderated; Comparation: habitual activities	Positive impact on caregivers’ subjective burden and risk of depression
Connell CM et Janevic NR, 2009 USA [26]	N = 137, female, 67 years old, wives	Intervention: telephone-based exercise; Duration: 6 months; Frequency: 30 min/session 3 session/week; Intensity: low to moderate; Comparation: not intervention	Reductions in perceived stress, increases in exercise self-efficacy
Kim et al., 2023 South Korea [27]	N = 64, female, 60 years old, relatives	Intervention: physical activity program; Duration: 8 weeks; Frequency: 60 min/session, 1 session/week; Intensity: not available; Comparation: not intervention	Improvements in self-efficacy, physical function and quality of life; decrease in caregiving burden and depression
Lök et al., 2023 Turkey [28]	N = 60, female, 50 years old, relatives	Intervention: Physical Activity Program; Duration: 8 weeks; Frequency: 1 session/week, 60 min/session; Intensity: moderated; Comparation: normal routine	Improvements in lifestyle and burden
Hives et al., 2021 Canada [29]	N = 68, most female, 61 years old, daughters or wives	Intervention: Aerobic exercise program; Duration: 24 weeks; Frequency: 3 sessions/week, 20 min/session; Intensity: in increase; Comparation: normal routine	Decrease in burden and in reported depression
Lin et al., 2023 USA [30]	N = 76, 92% female, 56 years old, most daughters	Intervention: social app to promote physical activity and well-being with daily step; Duration: 6 weeks; Frequency: daily; Intensity: light and moderate to vigorous; Comparation: app without social contact	Improvements in well-being and social support
Puterman et al., 2018 Canada [31]	N = 68, most female, 63 years old, daughters or spouses	Intervention: physical activity program; Duration: 24 weeks; Frequency: 40 min/session, 3–5 sessions/week; Intensity: Moderate to vigorous; Comparation: waitlist	Reduction in perceived stress and an increase in cardiorespiratory fitness
Castro et al., 2002 USA [32]	N = 100, female, 62 years old, relatives	Intervention: Home-based physical activity; Duration: 12 months; Frequency: 30 min/session, 4 sessions/week; Intensity: moderated; Comparation: Nutrition education	Improvement in perceived stress, burden, and depression
Yilmaz et al., 2019 Turkey [33]	N = 44, most female, 50 years old, relatives	Intervention: progressive muscle relaxation exercises at home; Duration: 8 weeks; Frequency: 28 min/day, 3 days/week; Intensity: not available; Comparation: nonintervention	Decrease in the caregiver burden and level of depression
Montero-Cuadrado et al., 2020 Spain [34]	N = 68, female, 64 years old, daughters or spouses	Intervention: family caregiver care program and physical exercise; Duration: 12 weeks; Frequency: 60 min/session, 3 sessions/week; Intensity: not available; Comparation: family caregiver care program	Improvement in quality of life, subjective burden, anxiety, depression and physical condition
Prieto-Prieto et al., 2022 Spain [35]	N= 48, female, 60 years old, most daughters	Intervention: Home-based physical exercise intervention; Duration: 9 months; Frequency: 60 min/session, 2 sessions/week; Intensity: moderate; Comparation: Normal daily activities.	Improvements in the dimensions of general health, vitality and mental health
King et al., 2002 USA [36]	N= 100, Female, 65 years old, relatives	Intervention: home-based, telephone-supervised exercise training; Duration: 12 months; Frequency: 30–40 min/session, 4 sessions/week; Intensity: moderate; Comparation: Nutrition education	Improvement in sleep quality and psychological distress
Flanagan et al., 2022 USA [37]	N= 32, most female, 57 years old, relatives	Intervention: Walking intervention; Duration: 8 weeks; Frequency: 30 min/day, 5 days/week; Intensity: not available; Comparation: Normal daily activity	Improvement in walked well-being and perceived stress
Hirano et al., 2011 Japan [38]	N= 31, most female, 74 years old, relatives	Intervention: Regular exercise; Duration: 12 weeks; Frequency: 3 sessions/week; Intensity: moderate; Comparation: Non exercise	Reduction in the burden and in the feeling of fatigue; improvement in quality of sleep
Farran et al., 2016 USA [39]	N= 211; most female; 61 years old; daughters or wives	Intervention: physical activity at home; Duration: 12 months; Frequency: ≥150 min/week; Intensity: moderate to vigorous; Comparison: caregiver skill building	Increase in physical activity and the number of steps maintained
Gary et al., 2020 USA [40]	N = 127, most female, 55 years old, spouse or other family	Intervention: Home-based aerobic and resistance exercise program and psychoeducation program; Duration: 6 months; Frequency: 4 sessions/week, 12 weeks; Intensity: low to moderate; Comparation: psychoeducation	Improvements in physical function and caregiver perception
Loi et al., 2024 Australia [41]	N = 121, 82% female, 70 years old, 78% spouse	Intervention: Home-based physical activity; Duration: 6 months; Frequency: 30 min/day, 5 days/week; Intensity: individualized for each patient; Comparation: Semi-structured discussions	Reduced depressive symptoms

## Data Availability

Data from the present investigation are available by contacting the corresponding author.

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
