# Peer review of "Effects of Physical Exercise on the Physical and Mental Health of Family Caregivers: A Systematic Review"

_healthcare, 2025, doi:10.3390/healthcare13101196_

Round 1
Reviewer 1 Report
Comments and Suggestions for Authors
Dear authors,
Thank you for the opportunity to review your manuscript. The study addresses a relevant topic and is generally well written; however, several key aspects must be rigorously addressed before the manuscript can be considered for publication. Below, you will find detailed comments and suggestions:
1. Keywords:
- It is recommended to avoid repeating terms already included in the title.
- Additionally, the use of standardized MeSH terms is strongly encouraged to enhance the visibility, searchability, and indexing of the article in scientific databases.
2. Introduction:
- The introduction is currently too brief and lacks theoretical depth. It is advisable to expand it by including:
- Previous evidence regarding the effects of physical exercise in other populations.
- A review of prior systematic reviews related to the topic addressed in this manuscript.
- The differences or novel contributions of this review compared to previous ones.
- A clear justification of the relevance and implications of this work for the scientific community and/or professionals in the field.
3. Methods:
- The PRISMA checklist is provided as supplementary material and should be followed to ensure methodological rigor and completeness.
- Several important methodological aspects are either missing or insufficiently described, which affects the transparency, rigor, and reproducibility of the study. Specifically:
- Please clarify how many reviewers were involved in each phase of the review process.
- Indicate whether complementary strategies (e.g., snowballing) were used to identify additional sources.
- State whether grey literature was included in the search.
- The inclusion and exclusion criteria section should be structured more clearly, following the PICOS framework (Population, Intervention, Comparison, Outcomes, Study design).
- The concepts of "risk of bias" and "methodological quality" must be clearly distinguished, as they are not synonymous. Accordingly:
- The risk of bias should continue to be assessed (e.g., using the ROB tool).
- A separate assessment of methodological quality should be included using an appropriate instrument, based on the type of included studies.
- More information should be provided regarding the data extraction process, including the tools used, the number of reviewers, and procedures for resolving discrepancies.
- Please explain why a meta-analysis was not conducted.
4. Results:
- Figure 1: Please use the standardized PRISMA flowchart available on the official PRISMA website.
- Table 1: The amount of text is excessive. Consider reformatting for clarity, define all abbreviations in the table footnote, and include information regarding the intensity of the interventions. If this information is not available in the original studies, please state this explicitly in the text.
- The title of section 3.2.2 appears in Spanish and should be corrected to maintain language consistency throughout the manuscript.
- Figures 2 and 3: These figures are of low visual quality. It is recommended to regenerate them using Review Manager (RevMan) to improve resolution and clarity.
- Again, we emphasize the importance of clearly distinguishing between “risk of bias” and “methodological quality.”
- In addition to reporting adverse events, dropout rates should also be included, as adherence is a crucial factor in the effectiveness of these types of interventions and may have influenced the results.
5. Discussion and Conclusions:
- Both sections require significant expansion. The discussion should critically analyze the main findings, place them in the context of existing literature, and reflect on their theoretical and practical implications.
- Specific future research directions should be included, along with a more detailed discussion of the limitations of the present study.
We believe the manuscript has potential, but it requires thorough revision and the implementation of substantial improvements before it can be reconsidered for possible publication.
Author Response
Response 1: Please see the attachment.

Reviewer 2 Report
Comments and Suggestions for Authors
I would like to appreciate your hard work in carrying out this systematic review.
Strengths of the study
- Precise classification and presentation of therapeutic interventions and related outcome measures, thereby providing a clear and comprehensive understanding of the concept.
General Concept Comments
- Although a systematic review of the literature was conducted, The PICOS framework was not considered, and the clinical research question remains undefined. A more detailed elaboration on these points is required and would significantly enhance the quality of the Methods section.
- There are ambiguities regarding the PRISMA Flow diagram that should necessarily be addressed.
- The study is solely based on RCTs. Elaborations on disregarding other types of related research (e.g., cohorts, case-controls, case reports, etc.) would be informative.
- The effects of therapeutic exercise have been assessed in numerous systematic reviews and meta-analyses. Justifying the focus on the specific population of family caregivers for dependent older adults will highlight the significance of this research.
Specific comments
- Page 1, Line 13: “The work” or ‘The adverse effects of the work”? Correction required.
- Page 1, Lines 31-33: It would be beneficial to provide detailed and updated statistics.
- Page 2, Line 50: It is recommended to mention related systematic reviews regarding the effects of therapeutic exercise on other population segments.
- Page 2, Line 51: This sentence implies that “physical activity” is a risk factor.
- Page 2, Line 59, Methods: For clarification, a clinical research question for this systematic review could be summarized and presented as supplementary material. The below template could be utilized.
|
|
Primary |
Secondary |
|
Objectives |
|
|
|
Questions |
|
|
- Page 2, Line 59, Methods: The PICOS framework could benefit from further elaboration regarding the inclusion and exclusion criteria. Additionally. The below template could be consulted.
|
|
Inclusion |
Exclusion |
|
Population |
|
|
|
Intervention |
|
|
|
Comparison |
|
|
|
Outcome |
|
|
|
Study Design |
|
|
- Page 3, Line 98, Figure 1: Inconsistencies regarding the reported numbers in the PRISMA Flow diagram. Also, this diagram may be presented in the Methods section.
- Page 4, Line 11, Table 1: Incorrect table format. Paper titles are not required; reference numbers should be provided; proper segmentation of the content; strengths; limitations; etc.
- Page 10, Lines 211-213: Observed heterogeneity in the interventions regarding the included studies is not necessarily a limitation to the current research.
Check for verb tenses, and maintain a balance between active and passive voice throughout the manuscript.
Author Response
Response 2: Please see the attachment

Round 2
Reviewer 1 Report
Comments and Suggestions for Authors
Thank you to the authors for responding in detail to my comments. A few final aspects to consider are as follows:
The authors have not provided the PRISMA checklist as supplementary material, as was requested in the previous round, or at least it has not been visible to me.
The research question and the objectives of the study should be presented at the end of the introduction in order to facilitate the reader’s understanding of the work, since, as you know, the research question serves as the guiding thread of the manuscript.
As the authors have not included any tables in the main body of the manuscript, and given that the table titled “Summary of the study characteristics and main results” helps provide an overview of the relevant publications, please include this table within the Results section of the manuscript, rather than as supplementary material.
Please review the use of abbreviations in the manuscript. Abbreviations should be defined only the first time they appear in the text and then used consistently throughout. A clear example is "RCT," which the authors define on several occasions.
Author Response
Response 1: Please see the attachment.
The checklist is uploaded as a file in the non published material section.

Reviewer 2 Report
Comments and Suggestions for Authors
Thank you for the great work in revising the manuscript.
Comments on the Quality of English LanguagePlease, double check for grammatical and typing errors.
For instance:
Line 161: Figure 1: "CINALH".
Line 279: In the study "analyzed".
Line 284: Madruga et al. [37] "go".
Line 290: In another study "analyzed of" Castro
....
Author Response
Response 1: Please see the attachment.
